# Rehabilitation Practitioners’ Perceptions of Optimal Sitting and Standing Posture in Men with Normal Weight and Obesity

**DOI:** 10.3390/bioengineering10020210

**Published:** 2023-02-06

**Authors:** Jiling Ye, Ziang Jiang, Shijie Chen, Rongshan Cheng, Lili Xu, Tsung-Yuan Tsai

**Affiliations:** 1Rehabilitation Department, Shanghai Ninth People’s Hospital, Shanghai Jiao Tong University School of Medicine, Shanghai 200011, China; 2Med-X Research Institute, School of Biomedical Engineering, Shanghai Jiao Tong University, Shanghai 200030, China; 3Clinical Research Unit, Shanghai Ninth People’s Hospital, Shanghai Jiao Tong University School of Medicine, Shanghai 200011, China

**Keywords:** postural health, rehabilitation practitioners, standing, sitting, back pain, BMI

## Abstract

The concepts of “optimal posture (OP)” and “harmful posture (HP)” are commonly used, and specific spinal postures can contribute to back pain. However, quantitative descriptions of optimal and harmful standing (StP) and sitting (SP) postures are currently lacking, particularly for different body mass indices (BMIs). Therefore, this study aimed to identify and quantify the OPs and HPs of StP and SP at different BMIs and investigate the attitudes and beliefs of rehabilitation practitioners toward OPs and HPs. Overall, 552 rehabilitation practitioners were recruited to participate in a questionnaire survey to select the optimal position from seven sitting and five standing postures for each BMI healthy volunteer. The segmental relationships of each posture were qualified using the Vicon software. For normal BMI, the physiotherapists chose two SPs (48.19% and 49.64%) and one StP (80.42%) as the OP. One sitting SP (83.7%) and two standing StPs (43.48% and 48.19%) were selected as optimal for obese BMI. All the most commonly selected OPs had an upright lordotic posture, while the postures with slouched spinal curves or forward head postures were almost all selected as HP. Additionally, 96.74% of participants considered education about optimal SP and StP to be “quite” or “very” important. The OP of the StP and SP postures was mostly based on the vertical alignment of gravity lines and sagittal balance. For obese people, the rehabilitation practitioners’ observations may be erroneous, and further physical examination is necessary. Rehabilitation practitioners generally believe that postural education is essential in clinical practice.

## 1. Introduction

The concepts of “postural health,” including “optimal posture” (OP) and “harmful posture” (HP), have been discussed in the healthcare professional community for many years [1,2]. The Posture Committee of the American Academy of Orthopedic Surgeons [1] defined optimal posture as “…the state of muscular and skeletal balance which protects the supporting structures of the body against injury or progressive deformity …” and poor posture as “…a faulty relationship of the various parts of the body which produces increased strain on the supporting structures …” Terms such as “protection” and “injury” have been used for many years. However, the lack of qualitative descriptions of optimal and harmful standing (StP) and sitting postures (SP) may influence healthcare providers’ perceptions of posture.

Kendall et al. [3] and Woodhull et al. [2] reported that the optimal StP involves a mid-range position of the pelvis, slight lumbar lordosis, and thoracic kyphosis, with the head in a balanced position. However, several curves fit the qualitative description. Therefore, there is controversy over the alignment of the standing lateral view [4]:The center of gravity should be anterior to the talus [2];The ear, shoulder, hip, knee, and talus should be perfectly aligned [5];Posterior parts of the head, back, and gluteal muscles should be vertically aligned [6].

There is also a lack of consensus on the optimal SP, and the optimal SP curve is described as follows:A lordotic spinal curve similar to standing [7];Lumbar lordosis without details regarding the thoracic and cervical curve [8];Lumbar lordosis and slight thoracic kyphosis [9];Flat lower thoracic and lumbar regions [10].

In contrast, HPs tend to be more uniform, such as the forward head [11] and spinal hyperflexion [12,13,14].

Clinically, rehabilitation practitioners teach patients with spinal pain to maintain good posture and avoid bad posture. However, there is limited evidence that any specific spinal posture can cause back pain. Before concluding whether and how spinal posture should be modified in people with spinal pain, it is crucial to understand the perceptions of OP and HP as well as the importance attached to OP and HP by rehabilitation practitioners. Based on existing evidence, it was hypothesized that physiotherapists would select which postures are optimal and harmful and that these would be justified based on the assumption that they are safe and natural. It is unclear whether the perceptions of posture are consistent for different body mass indices (BMIs) [15]. Therefore, Vicon was used to ensure postural unity and determine whether the visual assessment of rehabilitation practitioners was accurate and whether there would be selection bias due to different BMIs.

Utilizing a cohort based in China, this study aimed to investigate the perceptions of rehabilitation practitioners on the optimal and harmful StP and SP; whether the perceptions of posture are consistent in different BMIs; and how these rehabilitation practitioners describe and quantify the characteristics they associate with OP and HP.

## 2. Materials and Methods

### 2.1. Study Design

A 6-month cross-sectional design was used for an online survey conducted in China. This survey was designed to explore the perceptions of rehabilitation practitioners regarding optimal and harmful StP and SP with different BMI. Ethical approval was granted by the institutional ethics committee (SH9H-2021-TK12-1). Participants were notified that the survey completion would be accepted as providing informed consent.

### 2.2. Survey Design and Piloting

This study referred to the questionnaire design of Korakaki et al. [15], in which several modifications were made. The survey consisted of four sections, as follows:A closed question regarding the perceived importance of postural education in clinical practice (4-point Likert scale: 1 = not important at all, 4 = very important);A multiple-choice between seven SPs and five StPs in different BMI models from photographs to select at least one of each as “optimal” and “harmful” for postural education;Free text boxes for choice justification;Demographic information, including sex, age, level of educational qualifications, years of experience, clinical area of expertise and certifications, work location, and employment status, was obtained to explore whether these variables could explain any variation in the selection of OP and HP.

The percentage of multiple-choice options was the number of times an option was selected ÷ the number of valid answers, which indicates the proportion of the number of people who chose this option among all the participants who filled in their answers. Thus, the sum of percentage of multiple topics might have exceeded 100%.

Ten physiotherapists attempted to fill in the first draft of the questionnaire; the research group discussed and corrected the options with questions and ambiguities to reduce the deviation in the understanding of the questions in the questionnaire and finally came up with the final draft of the questionnaire with consistent understanding.

### 2.3. Image Collection of Standing and Sitting Postures

Images were captured using a digital camera (Canon EOS 550D) at a distance of 3 m from the model.

Two 23-year-old men, with BMI 21 kg/m^2^ and 32 kg/m^2^, who had no history of spinal pain and were flexible enough to assume a variety of StPs and SPs acted as models. A series of postures described in the literature and common in clinical observations were selected, including changes in the curvature of the cervical, thoracic, and lumbar vertebrae. A combination of postures was designed from the common “lazy posture” (pelvic tilt, thoracolumbar kyphosis, and head protrusion) to the excessive lordosis standing posture (pelvic tilt, thoracolumbar lordosis, and head retraction).

Each participant secured 16 infrared reflection markers that were used to characterize the head, trunk, and pelvis segmented by an experienced physical therapist (PT) (Figure 1 and Figure 2). Markers were applied to the bilateral lateral margins of the orbit, bilateral mastoid process, main protuberance of the forehead between the eyebrows (MPFE), C7, T5, T10, L3, left and right anterior superior iliac spine, and left and right posterior superior iliac spine to represent the head, torso, and pelvis segments. The selection and location of segments were based on the study by Korakakis et al. [16] and were derived to a certain extent.

### 2.4. Angle Analysis among Segments

Each participant maintained a trained and determined posture for 10 s, recorded using Vicon cameras for analysis. The data were reconstructed using Vicon Nexus (Vicon, 2.7.1, Oxford, UK), and the 3D position information of the reflective markers was imported into MATLAB (MATLAB, R2020a, The MathWorks, Natick, MA, USA) to reform segments and calculate angles.

According to a previous study on posture [16], this study defined the following parameters in the sagittal view to quantitatively evaluate the perception of each posture for the Chinese, as shown in Figure 3:Head Angle: The Euler angle between the vertical vector of the head segment and torso segment, reflecting the tilt of the head in the sagittal plane.Neck Angle: The angle of neck orientation in the global coordinate system. The neck orientation is the vector connecting the C7 marker to the center of lateral margins of the orbit (CLMO), reflecting the degree of neck flexion.Head Title Angle: The angle between the vector connecting CLMO to MPFE and the vertical axis in the global Cartesian coordinate system, reflecting the degree of the head in the coronal plane.Cervicothoracic Angle (CTA): The angle formed between the vector constructed from CLMO to C7 and the vector from C7 to T5, reflecting the degree of the head.Thoracic Angle: The angle between the vector connecting C7 and T5 and the vector connecting T5 and T10, describing the degree of thoracic kyphosis.Thoracolumbar Angle: The angle between the T5 to T10 vectors and the T10 to L3 vectors.Lumbar–sacrum Angle (LSA): The Angle between the T10 and L3 vectors and the L3 and S2 vectors, representing the lumbar lordosis angle.Pelvic Angle (PA): The angle between the line from the center of the anterior superior iliac spine and the center of the posterior superior iliac spine and its projection on the horizontal plane, reflecting the degree of inclination of the pelvis in the sagittal position.

### 2.5. Participants

No sample-size calculations were performed. To be eligible, respondents had to confirm that they were therapists and rehabilitation doctors and specify whether they were undergraduates or graduates.

### 2.6. Data Analysis

The final sample of 552 participants was included in the analysis. Counts and percentages were used to describe the characteristics and surveyed choices of the participants. The chi-square and Fisher’s exact tests were used to compare categorical variables and evaluate the associations. Moreover, we employed sets of logistic regression models to better understand the factors. To confirm the consistency of participants’ judgments of the StPs and SPs between obese and slim people, we used McNemar’s test to assess the difference. All statistical analyses were performed using SPSS version 26.0 (Chicago, IL, USA). All tests were two-tailed, and statistical significance was defined as *p* < 0.05.

## 3. Results

### 3.1. Participants and Demographics

Overall, 552 rehabilitation practitioners participated in the survey. Table 1 presents the demographic characteristics of the participants. The proportion of men and women was similar (51.81% and 48.19%, respectively). Most participants had a bachelor’s degree (81.52%). PT accounted for the majority (65.95%) of the specialties, and multiple choices were used in the PT subspecialty field, among which the musculoskeletal treatment direction was representative (63.19%), followed by the nerve direction (46.15%).

### 3.2. Sagittal Spinal Angles for Sitting and Standing Postures with Normal Weight and Obesity

Simultaneously, six Vicon cameras were used to collect the static three-dimensional positions of 16 infrared marker points on the models’ torso, with a sampling frequency of 100 Hz. After pretraining and guidance, the models maintained different postures for a period of time, during which 10 s of data collection was conducted to obtain eight angles of the body and spine. Table 2 and Table 3 show the specific angle data for normal weight and obesity, respectively.

### 3.3. The Choice of OP and HP with Different BMI

For normal BMI, the rehabilitation practitioners selected SP 1-4 and 1-5 (48.19% and 49.64%) and StP 1-4 (80.42%) as optimal. Appendix A shows the higher the workplace rank (odds ratio [OR] = 0.779, 95% confidence interval [CI]: 0.543–0.907, *p* < 0.01) and the younger the age (OR = 0.759, 95% CI: 0.582–0.989, *p* < 0.05), the more inclined the rehabilitation practitioners were to choose these two SP. However, the optimal choice for obese BMI was SP 2-4 (83.7%) and StP 2-3 (43.48%) and 2-5 (48.19%). Between normal and obese BMI, even under the same posture alignment on Vicon, rehabilitation practitioners chose different OP in sitting and standing. Only sports injury majors made the same choice of optimal StP and SP for different BMI (OR = 0.570, 95% CI: 0.080–0.745, *p* < 0.05). However, in the HP, the same SPs 1, 7 (93.12% and 65.58% in normal, 89.49% and 77.17% in obese) and StP 2 (83.7% in normal and 92.39% in obese) were selected for different BMI. This result reflects a lack of complete consensus on OP. Even in the same posture, observation errors due to obesity may cause difficulties in the positioning of posture. However, the most commonly selected postures all had some variation in upright lordotic sitting; in contrast, slouched spinal curves or forward head posture was selected as harmful.

### 3.4. Importance of Postural Education and Targeted Training

Education regarding optimal StP and SP was considered “considerably” or “very” important by 96.74% of participants, and only five people thought that posture education was not important at all. Additionally, Table 4 shows 27.17% of the rehabilitation practitioners selected “evaluate the posture of patients every time” and 38.77% selected “evaluate frequently.” These two items were significantly related to PTs’ musculoskeletal subspecialty, sports injuries, and children (*p* < 0.05). However, 29.71% of the participants selected “do not” evaluate the posture of patients. Notably, 77.9% of the medical personnel believed that patients should “often” and “every time” complete targeted training (38.41% and 39.49%, respectively) for posture problems, and PTs of musculoskeletal and sports injuries were more inclined to training posture (*p* < 0.05).

## 4. Discussion

This cross-sectional study investigated for the first time the choice of OP and HP, selection differences under different BMI, and the relevant basis of rehabilitation practitioners in the Chinese Mainland through online questionnaires. It was found that the vast majority of rehabilitation practitioners surveyed (96.74%) believed that spinal posture re-education is very important in clinical practice, which is consistent with the suggestion of the Association of Physiotherapists [17]. Although posture re-education may play a role in managing spinal pain in some patients, there is no substantial evidence to show the specific posture that causes spinal pain. Perhaps because of “conforms to the natural alignment of the spine” or based on ergonomics and biomechanics considerations, most rehabilitation practitioners believe that the trunk upright lumbar lordosis posture in the SP and StP is the most optimal, while the lazy posture or the head-forward posture is a HP, which is consistent with previous research [15].

However, general stereotyping of posture may be based more on clinical views than on experimental evidence. Roussouly et al. [18] divided four sagittal shapes of the spine using X-ray images and found that the change in the pelvic inclination angle affects the global spinal alignment, while the pelvic inclination angle can change with posture. Muyor et al. [19] compared a motion capture system with an X-ray radiograph image and observed that there was high consistency (ICC = 0.90) in evaluating the sagittal force line of the spine. Mousavi et al. [20] tested the reliability of repeated measurements with Vicon and showed excellent consistency in the pelvic tilt angle, lumbar lordosis, and thoracic kyphosis. Therefore, as a harmless, repeatable, and accurate assessment tool, Vicon can quantify various posture angles.

### 4.1. Clinical Significance of Spinal Angle

Kendall et al. [3] and Woodhull et al. [2] believe that the optimal StP involves a neutral pelvic position, mild lumbar kyphosis, and head in a balanced position, but no specific angle value was given; therefore, there is still much controversy about the optimal StP. Subsequent researchers [6,8,9,21] also studied the alignment arrangement of postures; only Korakakis et al. [15] used Vicon to quantitatively measure the angles of different postures, and the results were similar to those of this study; however, because the model in that study was a woman, there was still some deviation in the specific angle.

The increase in pelvic inclination angle may result in increased lumbar curvature or downward movement of curved vertices and lumbar curvature flattening [18]. Therefore, this study mainly used PA combined with LSA to describe the positions of the pelvis and lumbar vertebrae. PA represents the degree of pelvic tilt in the sagittal plane. The pelvic anteversion angle was defined as positive, and the posterior tilt angle was defined as negative. Lumbar lordosis was defined as positive and kyphosis as negative to prevent two different lumbar angle changes, combined with LSA to describe the lumbar lordosis angle. The combination of these two angles can show the curve of the entire pelvis and lumbar spine. Concurrently, CTA was used to describe the degree of the head forward, which is an important method to evaluate the upright torso [2]. The smaller the CTA value, the more serious the head protrusion. Table 2 and Table 3 show the other angle differences, which are no longer described in this study.

Upright lordotic postures are considered optimal despite the lack of strong evidence that any specific posture is linked to better health outcomes. While postural re-education may play a role in managing spinal pain in some patients, awareness of widespread and stereotypical beliefs regarding OP may be helpful in clinical assessment and management. For overweight or obese people, the observation of the physiotherapists may be incorrect, so further physical examination or quantitative equipment is needed for posture evaluation.

### 4.2. The Choice of Optimal Standing and Sitting Postures in Different Body Mass Index

In this study, the rehabilitation practitioners were asked to observe photographs of different postures to choose OP and HP. Thus, the observation process of the therapist in daily work was imitated. The results showed that most of the rehabilitation practitioners considered sitting posture 1-4 (CTA = 159.98°, LSA = 163.22°, PA = −10.52°), 1-5 (CTA = 155.96°, LSA = 163.22°, PA = −25.9°); and 2-4 (CTA = 159.02°, LSA = 168.99°, PA = 3.08°) of lumbar lordosis with the trunk upright as the OP, while 25.72% chose sitting posture 1-2 (CTA = 157.14°, LSA = 156.36°, PA = −1.40°) with the trunk upright and flat back.

However, some studies have pointed out that sitting positions 4 and 5 of lumbar lordosis activate more trunk muscles than the flat-back posture 2 [17,22,23], which may cause neck pain [24] and low back pain [25]. The long-term maintenance of the lumbar lordosis sitting posture is also questioned, as this state may exceed the bearing capacity of the paraspinal muscles [22], resulting in greater fatigue, pain, and discomfort in the neck and waist [26]. However, in a previous study [15], physiotherapists still chose the lumbar lordosis sitting posture in the vast majority (95.7%), which is consistent with our study.

Most of the rehabilitation practitioners chose standing posture 1-4 (CTA = 157.92°, LSA = 148.32°, PA = 9.93°), 2-3 (CTA = 162.74°, LSA = 153.83°, PA = 19.38°), and 2-5 (CTA = 165.73°, LSA = 153.51°, PA = 18.63°) as the optimal StP, mostly based on biomechanical models, such as vertical alignment of gravity lines [5,6], or “best” sagittal balance [18]. Roussouly et al. [18] recorded the changes in the sagittal plane of the standing spine through a prospective radiological study and divided the spinal curve into four categories:Lower lumbar segmental kyphosis and kyphosis in the thoracolumbar joint and thoracic vertebrae;Flatter spine, less thoracic kyphosis, and lumbar kyphosis;The inflection point of thoracic kyphosis, lumbar kyphosis at the thoracolumbar junction, and the peak of lumbar kyphosis at L4;Both the lumbar vertebrae and thoracolumbar joints are kyphotic.

Although the authors hypothesized that the type 3 spine is the “most balanced” posture, the correlation between spinal morphology and symptoms is unclear.

Some studies have proposed the use of a lumbar “neutral position” (defined as in the middle position and/or mild kyphosis) to relieve passive tissue tension in the terminal position, as to reduce potential pain [27] and facilitate the critical postural stabilizer muscle [22] and reduce the incidence of low back pain [28]. However, the neutral position is affected by age [29], lumbar mobility [17], and other factors. There is no consensus on whether the neutral position is actually upright or flexion.

### 4.3. The Choice of Harmful SP and StP with Different BMI

For the harmful SP, approximately 90% of the rehabilitation practitioners chose SP 1 of kyphosis and forward head 1-1 (CTA = 137.21°, LSA = 174.53°, PA = −14.39°) and 2-1 (CTA = 135.90°, LSA = 165.99°, PA = −12.80°). Moreover, 83.7% and 92.39% of the rehabilitation practitioners chose StP 1-2 (CTA = 155.52°, LSA = 153.69°, PA = 9.59°) and 2-2 (CTA = 157.49°, LSA = 154.76°, PA = 15.88°), respectively. However, some rehabilitation practitioners selected SPs 1-4 (12.32%) and 1-5 (12.68%) as HPs, although nearly half of the practitioners chose these two postures as OP, which is also consistent with the previous discussion on OP. It is believed that SP 4 and 5 may activate more muscles and cause spinal pain.

The choice of HP may reflect the long-standing view that flexion posture in the sitting position is related to increased intervertebral disc pressure [12,13], intervertebral disc creeps [14], and intervertebral disc degeneration. Nevertheless, the head protrusion position increases the load on the neck and fatigue of the neck extensor [16], which may cause neck pain [30]. Moreover, excessive lumbar kyphosis can excessively activate the paraspinal muscles [17,22,23], increasing facet joint pressure [18] and resulting in low back pain.

### 4.4. The Influence of Subspecialty on Posture-Related Choice

Rehabilitation practitioners generally believe that posture education is “quite” and “very” important in clinical practice. In all subspecialties, musculoskeletal, sports injuries, and children-oriented PT are more likely to evaluate the patient’s posture frequently, which may be due to the subprofessional’s need to assess whether the systemic or spinal axial symptoms of patients are related to a specific posture [15]. However, musculoskeletal and sports injury PTs are more inclined toward training posture, which may be related to the attributes of patients; these patients have no neurological abnormalities, most of them experience chronic pain, and they are often treated with static and dynamic postural habits [8,31].

Rehabilitation practitioners have a relatively uniform concept of HP; however, for optimal SP, professionals with younger and higher workplace levels were more inclined to choose sitting postures 4 and 5. This may be because the quality of medical care is proportional to the level of hospitals in China, and young practitioners are trained well by instructors in senior hospitals, resulting in a unified perception, or by associations of therapists [17].

Overall, rehabilitation practitioners tend to believe that OPs include an upright torso, mild lumbar kyphosis, and relative relaxation of the thoracic vertebrae based on the natural morphology of the spine and biomechanical principles and head retraction rather than kyphosis [23], though the “perfect posture” is a rare standard. Moreover, no specific normal posture or OP exists, and maintaining the same posture for a long time, no matter how ideal it is, will cause symptoms. Frequent changes in posture are required. Relatively, the concept of HP is the same, such as head protrusion and spinal flexion are obvious signs. Healthy people habitually assume a flexion spine posture in their work and life. Teaching patients to avoid such postures may be beneficial to some extent. Therefore, for clinical rehabilitation practitioners, it may be easier to unify and implement posture re-education to teach patients what constitutes a HP and to how to avoid it as well as to recommend frequent changes in posture.

### 4.5. Limitation

This study was limited to SPs and StPs of the static sagittal plane, and only a few spinal configurations were available to choose from, which could not include all postures. Therefore, reflecting on the dynamic changes in daily posture is difficult. The models used in the study were young men; however, differences in sex and age can also affect posture judgment. The rehabilitation practitioners involved in the survey were also young and had a low level of education and professional titles, which may have also resulted in some differences in the results. In addition, considering the small angle differences between some positions and changes in skin tissue adhesion and the surface of markers, unknown measurement errors may have occurred. For the quantification of sagittal spinal posture, CTA, LSA, and PA are more sensitive for identifying optimal and harmful spinal morphology, which can be further studied, providing a basis for follow-up research and development of simplified equipment. Further exploration is needed to assess whether postural feedback from wearable devices will affect the incidence and persistence of spinal pain.

## Figures and Tables

**Figure 1 bioengineering-10-00210-f001:**
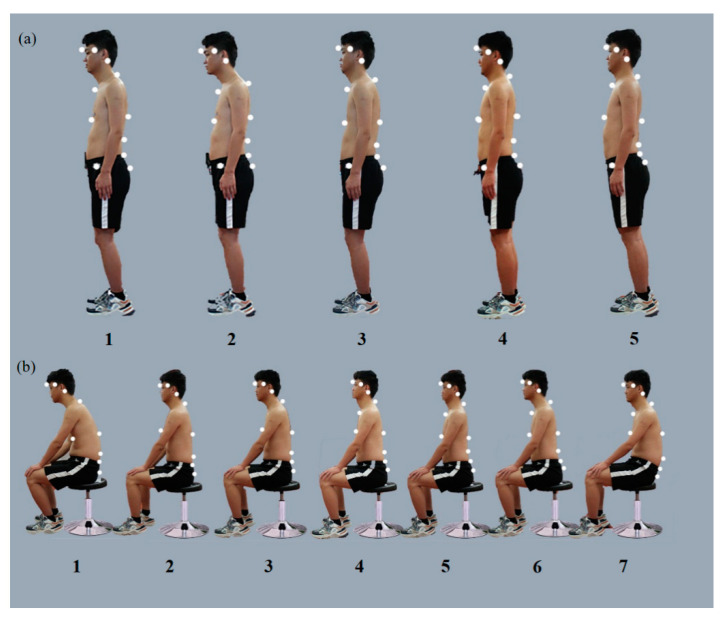
Sitting and standing postures of the first healthy volunteer with a BMI of 21 kg/m^2^: (**a**) the five determined standing postures; (**b**) the seven determined sitting postures.

**Figure 2 bioengineering-10-00210-f002:**
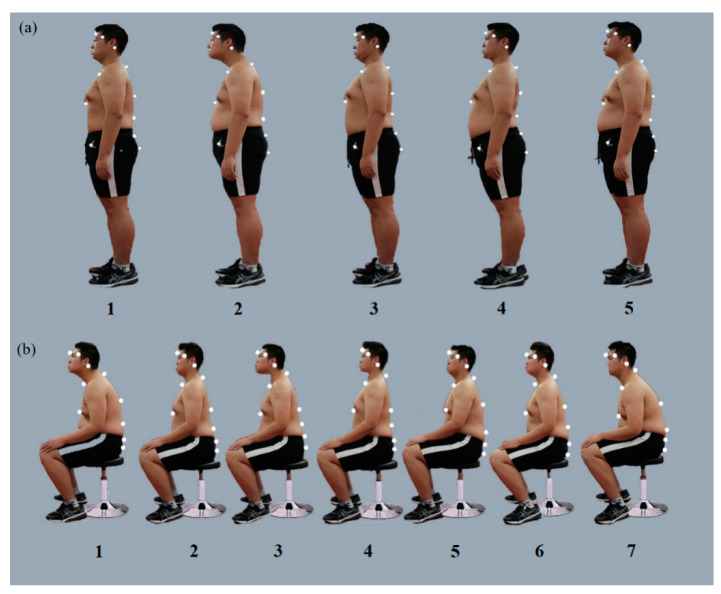
Sitting and standing postures of the second healthy volunteer with a BMI of 32 kg/m^2^: (**a**) the five determined standing postures; (**b**) the seven determined sitting postures.

**Figure 3 bioengineering-10-00210-f003:**
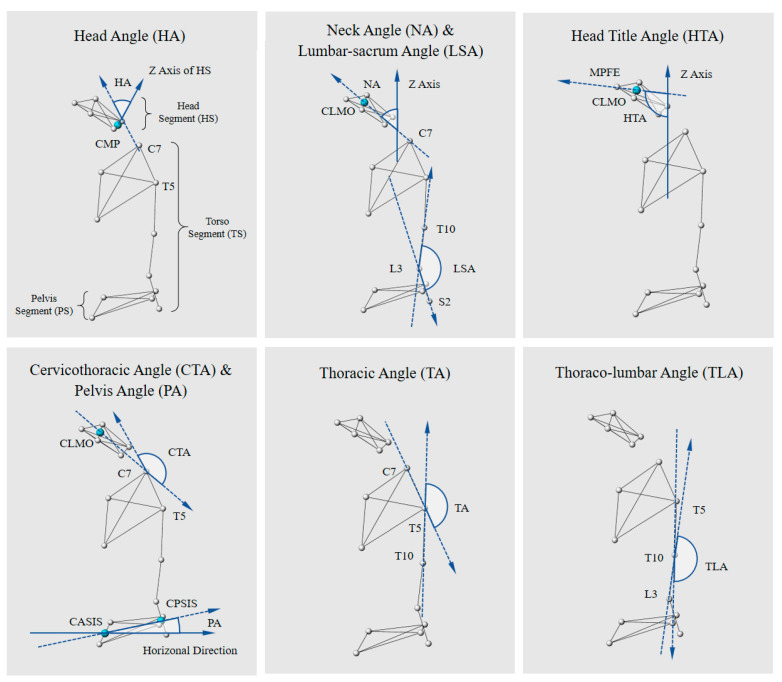
The eight sagittal angles calculated in the study describe the spinal configuration of each seated and standing posture. Abbreviations: CLMO, center of lateral margins of the orbit; CASIS, center of the anterior superior iliac spine; CPSIS, center of the posterior superior iliac spine; MPFE, main protuberance of the forehead between the eyebrows.

**Table 1 bioengineering-10-00210-t001:** Demographic and descriptive characteristics of the respondents (*n* = 552).

		(*n*)	%
Sex	Male	285	51.81
	Female	267	48.19
Age	18–25	151	27.54
	26–30	189	34.42
	31–40	173	31.16
	41–50	31	5.43
	51–60	8	1.45
Qualifications	Undergraduate	27	4.71
	Bachelor	449	81.52
	Master	64	11.59
	Doctor	12	2.17
Expertise	Rehabilitation doctor	77	13.95
	PT	365	66.12
	OT	33	5.98
	ST	22	3.99
	Others	55	9.96
Area of PT (multiple choices)	Musculoskeletal	236	63.19
	Neurological	194	46.15
	Cardiopulmonary	34	8.59
	Children	46	11.62
	Orthopedic	164	43.96
	Sports injuries	140	37.91
	Others	42	11.54
Years of expertise	0–3	175	31.88
Work status	3–6	119	21.38
	6–10	137	24.64
	>10	121	22.1
	Grade III hospital	293	53.26
	Grade II hospital	183	32.97
	Grade I hospital	19	3.62
	Self-employed	10	1.81
	Unemployed	21	3.62
	Others	26	4.71

Abbreviations: PT, physical therapist; OT, occupational therapist; ST, speech therapist.

**Table 2 bioengineering-10-00210-t002:** Sagittal spinal angles for sitting and standing postures of the first healthy volunteer with a BMI of 21 kg/m^2^.

	**Posture**	**HA(°)**	**NA(°)**	**HTA(°)**	**CTA(°)**	**TA(°)**	**TLA(°)**	**LSA(°)**	**PA(°)**
**STANDING**	1-1	74.9	57.9	103.4	149.1	157.1	154.7	153.8	7.1
	1-2	66.0	45.0	99.1	155.5	163.2	160.1	153.7	9.6
	1-3	62.5	36.9	100.7	161.1	166.4	160.6	152.8	9.4
	1-4	61.0	43.9	96.0	157.9	161.5	157.6	148.3	9.9
	1-5	64.2	45.7	97.1	155.7	162.8	160.1	152.4	10.3
	**Posture**	**HA(°)**	**NA(°)**	**HTA(°)**	**CTA(°)**	**TA(°)**	**TLA(°)**	**LSA(°)**	**PA(°)**
**SITTING**	1-1	86.5	60.5	102.8	137.2	164.7	158.5	174.5	−14.4
	1-2	61.7	48.0	94.0	155.2	169.5	169.7	170.9	−13.1
	1-3	70.1	61.1	96.3	147.5	163.6	166.1	171.4	−14.2
	1-4	57.2	43.4	94.2	160.0	168.7	169.7	163.2	−10.5
	1-5	62.6	45.7	95.7	156.0	169.4	165.3	176.4	−25.9
	1-6	62.1	49.1	93.6	154.5	168.4	164.9	175.9	−25.1
	1-7	69.0	52.9	95.9	149.5	164.3	162.9	175.2	−23.7

**Table 3 bioengineering-10-00210-t003:** Sagittal spinal angles for sitting and standing postures of the second healthy volunteer with a BMI of 32 kg/m^2^.

	**Posture**	**HA(°)**	**NA(°)**	**HTA(°)**	**CTA(°)**	**TA(°)**	**TLA(°)**	**LSA(°)**	**PA(°)**
**STANDING**	2-1	60.0	48.7	98.4	157.2	154.8	175.1	156.2	14.4
	2-2	57.2	54.4	97.8	157.5	153.4	177.3	154.8	15.9
	2-3	60.8	39.9	104.5	162.7	158.4	179.2	153.8	19.4
	2-4	56.1	42.8	99.2	163.6	160.1	177.9	156.1	15.8
	2-5	57.5	38.1	103.6	165.7	158.3	177.1	153.5	18.6
	**Posture**	**HA(°)**	**NA(°)**	**HTA(°)**	**CTA(°)**	**TA(°)**	**TLA(°)**	**LSA(°)**	**PA(°)**
**SITTING**	2-1	74.7	61.5	91.4	135.9	162.3	162.8	166.0	−12.8
	2-2	65.9	44.8	104.7	157.1	162.7	175.6	156.4	−11.4
	2-3	67.0	49.9	101.3	153.0	159.0	174.2	155.0	−10.6
	2-4	70.2	40.6	111.2	159.0	163.0	177.4	169.0	3.08
	2-5	57.7	38.3	101.8	165.1	164.1	176.7	164.2	5.09
	2-6	64.5	47.0	105.2	157.9	160.7	178.2	166.6	5.5
	2-7	67.2	48.0	99.7	151.9	164.9	166.5	164.6	−23.5

Abbreviations: HA, Head Angle; NA, Neck Angle; HTA, Head Title Angle; CTA, Cervicothoracic Angle; TA, Thoracic Angle; TLA, Thoracolumbar Angle; LSA, Lumbar–sacrum Angle; PA, Pelvic Angle.

**Table 4 bioengineering-10-00210-t004:** Number of people selected by PT subspecialties for Likert score of posture (*n*, %).

Subspecialty	Frequency of Postural Assessment					Frequency of Postural Training				
	No	Sometimes	Often	Every Time		No	Sometimes	Often	Every Time	
**Musculoskeletal**	6(2.5)	52(22.0)	91(39.0)	87(36.5)	*	3(0.8)	31(13.6)	95(39.8)	107(45.8)	#
**Neurological**	8(4.1)	69(35.5)	67(34.6)	49(25.8)		8(4.1)	30(15.5)	69(35.1)	87(45.3)	
**Cardiopulmonary**	0(0)	8(23.5)	12(35.3)	14(41.2)		0	4(11.8)	6(17.7)	23(70.5)	
**Children**	2(4.4)	3(6.6)	24(52.2)	17(34.8)	#	1(2.1)	6(13.2)	14(30.8)	25(53.9)	
**Orthopedic**	2(1.2)	36(21.6)	68(40.5)	58(36.7)		3(1)	21(13)	61(37)	79(49)	
**Sports injuries**	0	23(17.1)	62(44.3)	55(38.6)	*	0	16(11.4)	50(35.8)	74(52.8)	#
**Others**	1(1)	16(38.1)	13(33.2)	12(28.6)		1(2.4)	6(14.1)	18(42.6)	17(40.9)	

Fisher’s exact test, * *p* < 0.01, # *p* < 0.05.

## Data Availability

Not applicable.

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
