# Peer review of "Rehabilitation Practitioners’ Perceptions of Optimal Sitting and Standing Posture in Men with Normal Weight and Obesity"

_bioengineering, 2023, doi:10.3390/bioengineering10020210_

Round 1

Reviewer 1 Report

The paper shows a lack of consensus in OP selection. The definition of OP is not provided in the paper of course as it is discussed based on general opinion of rehabilitants based on their practice. No quantitative measures causing e. g. back pain are discussed in the paper. Thus, the paper provides just a general insight onto posture – healthy and unhealthy, however proves the above mentioned lack of consensus, which may be motivation to better educate rehabilitants or determine one definition of OP for each anthropometric group.

I have some critical remarks, which should be considered in the corrected manuscript:

1.      Line 18 “ Therefore, this study aimed to identify and quantify the OPs 18 and HPs of StP and SP at different BMIs”. In my opinion the Authors do not quantify optimal posture because the criteria for optimization are not specified. The criteria should refer e. g. to back pain level or pain occurrence in a certain population and not on subjective opinion of rehabilitants. 

2.      The criteria of ‘optimal’ and ‘harmful’ postures are not defined in the research. That is why I have a doubt concerning the objectivity of the research – in general each rehabilitant can have different idea on ‘optimal’ and ‘harmful’ posture

3.      All results of the survey should be shown in a table, for clarity. Discussion of just selected results is not enough to fully understand results of the study – I mean, it is important to see also how votes are distributed between different positions – evenly, or somehow different? That would show how certain a given result is and comment on that should also appear in the discussion.

4.      The Authors wrongly refer to literature, e. g. there should be Kendall et al. [3]and Woodhull et al.  [2] in line 45. Please check also other citations

5.      Line 198: “SPs between fat and thin people,” – I’d suggest words ‘obese’ and ‘slim’

6.      Line 98: for clearance, there should be “the SUM OF percentage of multiple topics might have exceeded 100%.”

7.      Please specify unit of angles placed in Tables 2 and 3

8.      I’d suggest showing 1 number after dot in angles shown in Tab. 2, 3 

Author Response

Reply to Reviewer 1s Comments

 Reviewer 1’s comments:

The paper shows a lack of consensus in OP selection. The definition of OP is not provided in the paper of course as it is discussed based on general opinion of rehabilitants based on their practice. No quantitative measures causing e. g. back pain are discussed in the paper. Thus, the paper provides just a general insight onto posture-healthy and unhealthy, however proves the above mentioned lack of consensus, which may be motivation to better educate rehabilitants or determine one definition of OP for each anthropometric group.

Reply: Thank you very much for reviewing our manuscript. We appreciate your recognization. Your suggestions are constructive and beneficial to improving the paper. The earliest definition of optimal posture comes from the AAOS in 1947. Later experts, such as Kendall, Woodhull, and McKenzie, described the posture but did not quantify it. So we want to quantify the value of this posture. However, because there are too many causes of low back pain, posture may be a big factor, and the quantitative posture of patients with low back pain is also the next direction for us to work hard. And about the language problem, we have invited a native speaker to revise it, however, limited by the deadline we can only handle the improved manuscript later.

Question 1 - Line 18 "Therefore, this study aimed to identify and quantify the OPs 18 and HPs of StP and SP at different BMIs". In my opinion, the Authors do not quantify optimal posture because the criteria for optimization are not specified. The criteria should refer e. g. to back pain level or pain occurrence in a certain population and not to the subjective opinion of rehabilitates.

Reply: Thank you very much for your suggestion. At present, there is only one definition of OP and HP. But in clinical practice, it is not clear which posture is more consistent with the definition of OP and HP, especially those with different BMI. Therefore, one of the main purposes of this paper is to determine which posture is more in line with the previous definition through the choice of rehabilitates. And make a follow-up quantification on this choice..

 Question 2 - The criteria of optimal and harmful postures are not defined in the research. That is why I have a doubt concerning the objectivity of the research in general each rehabilitant can have different idea on optimal and harmful posture

Reply: We very much agree with your point of view, so in the process of experimental design, first of all, we draw lessons from the previous article to adjust the 5 kinds of standing posture and 7 kinds of sitting posture of the subjects. secondly, we try to include as many rehabilitates as possible to conduct a questionnaire survey to ensure reliability. We have enhanced this part of the narrative in this version.

Question 3 - All results of the survey should be shown in a table, for clarity. Discussion of just selected results is not enough to fully understand results of the study I mean, it is important to see also how votes are distributed between different positions evenly, or somehow different? That would show how certain a given result is and comment on that should also appear in the discussion.

Reply: Thank you for your comment. Because the content of the questionnaire is very large, there is no obvious difference except for age, workplace, professional orientation, etc. I indicated in the text, that the content of the questionnaire had not been completely displayed before. Accordingly, we will mark all the data in the form of an appendix at the end of the article.

Question 4 - The Authors wrongly refer to literature, e. g. there should be Kendall et al. [3]and Woodhull et al. [2] in line 45. Please check also other citations.

Reply: Thank you very much for your advice. We apologize for these mistakes and make the following revise accordingly. And we also checked to make sure no other mistakes in other citations.

Line 46-47: Kendall et al. [3]and Woodhull et al. [2] reported that the optimal StP involves a mid-range position of the pelvis, slight lumbar lordosis, and thoracic kyphosis, with the head in a balanced position.

Line 86-87: This study referred to the questionnaire design of Korakaki et al.[15], in which several modifications were made. The survey consisted of four sections as follows:......

Line 292-293: Kendall et al. [3] and Woodhull et al. [2] believe that the optimal StP involves a neutral pelvic position......

 Question 5 - Line 198: "SPs between fat and thin people," Id suggest words obese and slim

Reply: Thank you very much for your advice. We revised it and the article had become more rigorous.

Line 119-120: To confirm the consistency of participants’ judgments of the StPs and SPs between obese and slim people......

 Question 6 - Line 98: for clearance, there should be "the SUM OF percentage of multiple topics might have exceeded 100%."

Reply: Thank you very much for your advice. We revised it and the article had become more rigorous.

Line 100-101: Thus, the sum of the percentage of multiple topics might have exceeded 100%.

 Question 7 - Please specify unit of angles placed in Tables 2 and 3.

Reply: Thank you very much for your advice. We add the unit accordingly and apologize for the mistake.

 Question 8 -  Id suggest showing 1 number after dot in angles shown in Tab. 2, 3

Reply: Thank you very much for your advice. We adjusted the number accordingly in this version.

Reviewer 2 Report

General comments

This manuscript aims at investigate 1. the perceptions of rehabilitation practitioners on the optimal and harmful standing and sitting postures, 2. whether the perceptions of posture are consistent across different body mass indices and 3. how these rehabilitation practitioners describe and quantify the characteristics they associate with optimal and harmful postures. Several specific and minor issues detailed below prevent to recommend acceptance for publication.

Specific comments

Why were no sample-size calculations performed?

I do not believe in the straight equation BMI=21 kg/m2->normal and BMI=32 kg/m2->obese. At least, fat mass should be assessed with some literature skinfold-based equation.

Now, there is a comparison between rehabilitation experts and motion analysis assessments. Yet, different scoring ranges should be disclosed. I mean, e.g., if variable X results from n to m, then motion analysis assessment says “very high upright lordotic posture”. Then, rehabilitation experts assessment should be compared with motion analysis. Otherwise, reader gets confused information.

Discussion could benefit from some figures showing different postures distributions in terms of different participants absolute values and/or percentages choosing one posture instead of others as optimal or harmful.

Minor comments

(line 22) … for each BMI subject. ?

(l25) … All the most commonly selected OPs had…

(l45 and elsewhere throughout MS, regarding need for adding “et al.” to in-text citations) … Kendall et al. [3] and Woodhull et al. …

(l104 and elsewhere throughout MS) please, do not use acronyms in headings;

(l115) … therapist (PT) (Figures…

(l144) … (CLMO), reflecting…

(l149) … reflecting the degree of the head. ?

(l193) … we analyzed the data regardless. ??

(l298) please, do not start sentences with acronyms.

Author Response

Reply to Reviewer 3s Comments

Reviewer 3’s comments:

Thank you for the privilege of reviewing the manuscript describing a research study into rehabilitation practitioners' perceptions of optimal sitting and standing posture in men with normal weight and obesity.

The observation that rehabilitation practitioners with sub-specialization in musculoskeletal, sports injuries, and children-oriented PT are more likely to evaluate the patient's posture frequently is an important finding. I agree with the authors (line 398) that reflecting on the dynamic changes of daily posture is difficult.

The main shortcoming of this study is in my opinion that the authors only used images of 2 male actors and did not include images of two female actors. Gender bias will inevitably occur as a consequence and the findings of authors thus cannot be generalized. This shortcoming in study design hampers the scientific value of this investigation.

Reply: Thank you very much for reviewing our manuscript, and we are encouraged by your review advice. Our study quantifies postural control, which tends to have similar effects across genders, and has been described in previous studies on postural adjustment in female samples[1]. In this study, we do not discuss female and male comparisons directly, but we have added descriptions in the discussion and limitations section and look forward to future opportunities to do further research. Your suggestions are constructive and beneficial to improving the paper. We have revised the manuscript accordingly. We hope this manuscript is now in a form suitable for publication. And about the language problem, we have invited a native speaker to revise it, however, limited by the deadline we can only handle the improved manuscript later.

[1] Korakakis V, O'Sullivan K, O'Sullivan P B, et al. Physiotherapist perceptions of optimal sitting and standing posture[J]. Musculoskeletal Science and Practice, 2019, 39: 24-31.

Question 1 - Please include in the title: "in men" (replace: "... standing posture ..." by " ... standing posture in men ..."

Reply: Thank you very much for your suggestion. We have added “in men” into our title. 

Line2-3: Rehabilitation practitioners’ perceptions of optimal sitting and standing posture in men with normal weight and obesity

Question 2 - Please provide percentages together with numbers in Tab.4.

Reply: Thank you for your comment. We add percentages with numbers in Tab.4.

Question 3 - Please be humble and replace "subjects" by "patients" or "healthy volunteers" or another word.

Reply: Thank you very much for your careful review. We adjusted the number accordingly in this version.

Line 225-226: Table 3. Sagittal spinal angles for sitting and standing postures of the second healthy volunteer with a BMI of 32 kg/m2.

Line 221-222: Sagittal spinal angles for sitting and standing postures of the first healthy volunteer with a BMI of 21 kg/m2.

Line 131-132: Figure 2. Sitting and standing postures of the second healthy volunteer with a BMI of 32 kg/m2. (a) The five determined standing postures; (b) the seven determined sitting postures.

Line 128-129: Figure 1. Sitting and standing postures of the first healthy volunteer with a BMI of 21 kg/m2. (a) The five determined standing postures; (b) the seven determined sitting postures.

Line 20-22: Overall, 552 rehabilitation practitioners were recruited to participate in a questionnaire survey to select the optimal position from seven sitting and five standing postures for each BMI-healthy volunteer.

Question 4 - page 4 table 1: please explain "tor"

Reply: Because of the table problem, the content is “Doctor”, which is shown as “Doc-tor”. We have fixed the problem. We have revised it.

Question 5 - page 5 table 1 and page 9 Table 4 : replace "Orthopedical" by "Orthopedic"

Reply: Thank you very much for your advice. We replace the word in both Table 1 and Table 4.

Question 6 - page 5 table 1: please check 34 (9.34%) cardiopulmonary PT and 46 (9.34%) PT treating children.  Both groups represent 9.34%, is this correct?

Reply: Thank you very much for your advice. We revised the 46 (9.34%) to 46 (11.62%), 34 (9.34%) to 34 (8.59%), and felt sorry for the mistake.

Question 7 - Line 288: replace: "believe" by "hypothesize".

Reply: Thank you very much for your advice. We make the revision accordingly.

Line 342-343: Although the authors hypothesized that the type 3 spine is the "most balanced" posture, the correlation between spinal morphology and symptoms is unclear.

Question 8 - replace "is given" by "was given".

Reply: Thank you very much for your advice. We make the revision accordingly.

Line 286-288: Kendall et al. [3] and Woodhull et al. [2] believe that the optimal StP involves a neutral pelvic position, mild lumbar kyphosis, and head in a balanced position, but no specific angle value was given

Reviewer 3 Report

Thank you for the privilege of reviewing the manuscript describing a research study into the rehabilitation practitioners' perceptions of optimal sitting and standing posture in men with normal weight and obesity.

Please include in the title: "in men" (replace: "... standing posture ..." by " ... standing posture in men ..."

The observation that rehabilitation practitioners with sub-specialization in musculoskeletal, sports injuries, and children-oriented PT are more likely to evaluate the patients' posture frequently is an important finding.

Please provide percentages together with numbers in Tab.4.

Please be humble and replace "subjects" by "patients" or "healthy volunteers" or another word.

page 4 table 1: please explain "tor"

page 5 table 1 and page 9 Table 4 : replace "Orthopedical" by "Orthopedic"

page 5 table 1: please check 34 (9.34%) cardiopulmonary PT and 46 (9.34%) PT treating children.  Both groups represent 9.34%, is this correct? 

Line 288: replace: "believe" by "hypothesize".

Line 290: replace "is given" by "was given".

I agree with authors (line 398) that reflecting on the dynamic changes of daily posture is difficult.

The main shortcoming of this study is in my opinion that the authors only used images of 2 male actors and did not include images of two female actors. Gender bias will inevitably occur as a consequence and the findings of authors thus cannot be generalized. This shortcoming in study design hampers the scientific value of this investigation.

Author Response

Reply to Reviewer 3s Comments

Reviewer 3’s comments:

Thank you for the privilege of reviewing the manuscript describing a research study into the rehabilitation practitioners' perceptions of optimal sitting and standing posture in men with normal weight and obesity.

The observation that rehabilitation practitioners with sub-specialization in musculoskeletal, sports injuries, and children-oriented PT are more likely to evaluate the patients' posture frequently is an important finding. I agree with authors (line 398) that reflecting on the dynamic changes of daily posture is difficult.

The main shortcoming of this study is in my opinion that the authors only used images of 2 male actors and did not include images of two female actors. Gender bias will inevitably occur as a consequence and the findings of authors thus cannot be generalized. This shortcoming in study design hampers the scientific value of this investigation.

Reply: Thank you very much for reviewing our manuscript, and we are encouraged by your review advice. Our study quantifies postural control, which tends to have similar effects across genders, and has been described in previous studies on postural adjustment in female samples[1]. In this study, we do not discuss female and male comparisons directly, but we have added descriptions in the discussion and limitations section and look forward to future opportunities to do further research. Your suggestions are constructive and beneficial to improving the paper. We have revised the manuscript accordingly. We hope this manuscript is now in a form suitable for publication. And about the language problem, we have invited native speaker to revise it, however, limited by the deadline we can only handle the improved manuscript later .

[1] Korakakis V, O'Sullivan K, O'Sullivan P B, et al. Physiotherapist perceptions of optimal sitting and standing posture[J]. Musculoskeletal Science and Practice, 2019, 39: 24-31.

Question 1 - Please include in the title: "in men" (replace: "... standing posture ..." by " ... standing posture in men ..."

Reply: Thank you very much for your suggestion. We have added “in men” into our title. 

Line2-3: Rehabilitation practitioners’ perceptions of optimal sitting and standing posture in men with normal weight and obesity

Question 2 - Please provide percentages together with numbers in Tab.4.

Reply: Thank you for your comment. We add percentages with numbers in Tab.4.

Question 3 - Please be humble and replace "subjects" by "patients" or "healthy volunteers" or another word.

Reply: Thank you very much for your careful review. We adjusted the number accordingly in this version.

Line 225-226: Table 3. Sagittal spinal angles for sitting and standing postures of the second healthy volunteer with a BMI of 32 kg/m2.

Line 221-222: Sagittal spinal angles for sitting and standing postures of the first healthy volunteer with a BMI of 21 kg/m2.

Line 131-132: Figure 2. Sitting and standing postures of the second healthy volunteer with a BMI of 32 kg/m2. (a) The five determined standing postures; (b) the seven determined sitting postures.

Line 128-129: Figure 1. Sitting and standing postures of the first healthy volunteer with a BMI of 21 kg/m2. (a) The five determined standing postures; (b) the seven determined sitting postures.

Line 20-22: Overall, 552 rehabilitation practitioners were recruited to participate in a questionnaire survey to select the optimal position from seven sitting and five standing postures for each BMI healthy volunteer.

Question 4 - page 4 table 1: please explain "tor"

Reply: Because of the table problem, the content is “Doctor”, which is show as “Doc-tor”. We have fixed the problem.

Question 5 - page 5 table 1 and page 9 Table 4 : replace "Orthopedical" by "Orthopedic"

Reply: Thank you very much for your advice. We replace the word in both Table 1 and Table 4.

Question 6 - page 5 table 1: please check 34 (9.34%) cardiopulmonary PT and 46 (9.34%) PT treating children.  Both groups represent 9.34%, is this correct?

Reply: Thank you very much for your advice. We revised the 46 (9.34%) to 46 (11.62%), 34 (9.34%) to 34 (8.59%) and felt sorry for the mistake.

Question 7 - Line 288: replace: "believe" by "hypothesize".

Reply: Thank you very much for your advice. We make the revise accordingly.

Line 342-343: Although the authors hypothesized that the type 3 spine is the "most balanced" posture, the correlation between spinal morphology and symptoms is unclear.

Question 8 - replace "is given" by "was given".

Reply: Thank you very much for your advice. We make the revise accordingly.

Line 286-288: Kendall et al. [3] and Woodhull et al. [2] believe that the optimal StP involves a neutral pelvic position, mild lumbar kyphosis, and head in a balanced position, but no specific angle value was given

Round 2

Reviewer 2 Report

General comments

Authors addressed my issues well enough.

Reviewer 3 Report

Thank you for revision of the manuscript. You improved the manuscript.